# A Novel Cu^2+^ Quantitative Detection Nucleic Acid Biosensors Based on DNAzyme and “Blocker” Beacon

**DOI:** 10.3390/foods12071504

**Published:** 2023-04-03

**Authors:** Hanyue Zhang, Kai Dong, Shuna Xiang, Yingting Lin, Xiaoyan Cha, Ying Shang, Wentao Xu

**Affiliations:** 1Faculty of Food Science and Engineering, Kunming University of Science and Technology, Kunming 650500, China; 2College of Biological Sciences, China Agricultural University, Beijing 100083, China; 3Key Laboratory of Precision Nutrition and Food Quality, Department of Nutrition and Health, China Agricultural University, Beijing 100083, China

**Keywords:** biosensor, Cu^2+^, DNAzyme, ligation, PCR, G-quadruplex

## Abstract

In this paper, a “turn-off” biosensor for detecting copper (II) ions based on Cu^2+^-dependent DNAzyme and a “blocker” beacon were developed. Upon the copper ion being added, the Cu^2+^-dependent DNAzyme substrate strand was irreversibly cleaved, thereby blocking the occurrence of the ligation reaction and PCR, which inhibited the G-rich sequence from forming the G-quadruplex structure, efficiently reducing the detection signal. This method had the characteristics of strong specificity and high sensitivity compared with the existing method due to the application of ligation-dependent probe signal recognition and amplification procedures. Under the optimized conditions, this method proved to be highly sensitive. The signal decreased as the concentration of copper ions increased, exhibiting a linear calibration from 0.03125 μM to 0.5 μM and a limit of detection of 18.25 nM. Subsequently, the selectivity of this biosensor was verified to be excellent by testing different relevant metal ions. Furthermore, this detection system of copper (II) ions was successfully applied to monitor Cu^2+^ contained in actual water samples, which demonstrated the feasibility of the biosensor.

## 1. Introduction

In the environment, heavy metal ions naturally exist in the atmosphere, oceans, and soils due to geological causes. With the progress of human society, mining and smelting in industrial production activities, the use of heavy metal ion pesticides and cadmium-containing fertilizers in agricultural production activities, waste incineration, and automobile exhaust emissions in daily life activities have made heavy metal pollution increasingly serious [1]. These heavy metal ions enter the water and soil and enter the food chain via plants, animals, and microorganisms. Heavy metal ions are absorbed by plants through their roots, transmitted from stems to other tissues, and then accumulated. For plants themselves, the presence of heavy metal ions affects their own growth [2,3]. Animals need to ingest water and other foods for survival. In this process, heavy metal ions may accumulate in animals. In this way, heavy metal ions in the food chain can be transmitted between various nutritional levels, and bioaccumulation, transformation, and biomagnification will eventually endanger human health [4,5]. The Minamata disease caused by mercury-containing industrial wastewater discharge pollution and the painful Itai-itai disease caused by cadmium rice produced on cadmium-containing water-irrigated farmland are warning us to pay attention to the health problems caused by heavy metal pollution.

Nowadays, more and more people pay attention to heavy metal pollution in food safety. Heavy metal ions are a kind of chemical substance that has obvious toxicity to organisms [6]. Heavy metal ions may cause kidney damage, lung damage, bone damage, nervous system damage, etc. [7]. The heavy metal ions are difficult to degrade by organisms in nature; however, they are widely distributed and then harmful to the environment and human health [8,9]. The copper ion is one of the most important ones among these heavy metal ions. It is a necessary trace element for human growth, and only a small amount of copper ions is needed to maintain the normal life activities of the human body. However, excessive copper ions will interfere with the normal physiological function of the human body and further endanger human health. For example, the gradual accumulation of copper ions in the liver will cause damage to it, and copper can catalyze the formation of hydroxyl radicals, resulting in DNA damage and increased lipid peroxidation [10]. Therefore, it is significant to detect copper ions effectively and sensitively.

There are many ways to detect copper ions, such as the traditional detection methods based on large instruments, including atomic absorption spectrometry (AAS) [11], gas chromatography (GC), inductively coupled plasma mass spectrometry (ICP-MS), high performance liquid chromatography (HPLC) [12,13], and so on. These methods rely on relevant large-scale equipment, expensive supplies, and standard substances, which have the merits of high sensitivity and good specificity. However, the above methods are also inadequate sometimes because of complex sample preparation, a long detection period, and high costs. Hence, it is difficult to meet the requirements of the modern fast, simple, and high-throughput detection tendency. 

In recent years, functional nucleic acids have shown great potential and developed rapidly in the detection of heavy metal ions. It has been applied to the detection of many kinds of heavy metal ions, including Pb^2+^ [14,15], Hg^2+^ [16,17], Ag^+^ [18,19], Cu^2+^ [20,21,22] and UO_2_^2+^ [23,24]. The term functional nucleic acid refers to a single-chain DNA fragment with catalytic function that has specific structure recognition capacity and excellent catalytic activity. It includes metal ion-dependent DNAzymes, and as a DNA-based biocatalyst, it can carry out many biological and chemical reactions. Refs. [25,26] exhibit catalytic activity only when specific metal ions exist and can cause DNA double-strand fragments to break at specific positions to produce smaller oligonucleotide fragments.

A large number of colorimetric [26,27,28,29,30,31], electrochemical [32,33,34], and fluorescent [21,22,35,36,37,38,39,40,41,42,43,44] DNAzyme-based biosensors have been designed to determine Cu^2+^ with high sensitivity and selectivity. Compared with electrochemical and fluorescent DNAzyme sensors, colorimetric sensor technology does not require the use of more expensive instruments and is low cost, which can achieve fast, sensitive, on-site, and visual inspection. At present, the colorimetric sensor technology based on functional nucleic acid detection of copper ions can be divided into two major categories according to gold nanoparticles and G-quadruplex. Wang established a label-free colorimetric method for the detection of metal ions by combining Cu^2+^-dependent DNAzyme with the color change characteristics of gold nanoparticles after aggregation. Under the cutting action of copper ions, the substrate chain of DNAzyme is cut and ssDNA is released. They are adsorbed on gold nanoparticles so that gold nanoparticles cannot aggregate and the system does not change color [45].

G-quadruplex DNAzymes have seen rapid development in colorimetric sensor design in the past period of time. The G-quadruplex structure is the secondary structure of DNA.

Hemin is used as a cofactor for many oxidases. When hemin is inserted into the G-quadruplex, the catalytic activity of the compound is enhanced [46,47]. Studies have shown that the G-quadruplex was used as a new type of DNAzyme with peroxidase activity, the complexes of which were formed by hemin and can catalyze the oxidation of 2,2′-Azino-bis (3-ethylbenzthiazoline-6-sulfonic acid) (ABTS) to produce the free radical cation ABTS^+^ in the presence of H_2_O_2_, and the absorption signal increases at ~419 nM [48,49]. Yin realized the highly sensitive detection of copper ions by combining the Cu^2+^-dependent DNAzyme, substrate, and an HRP-mimicking DNAzyme into a unimolecular colorimetric sensor [50]. Zhang used the substrate strand of the DNA-cleaving DNAzyme and a G-rich sequence as a stem-loop structure, and in the presence of copper ions, the substrate chain cleavage, which can release the G-rich sequence and subsequently form the G-quadruplex DNAzyme, detected the copper ion [51]. For biosensor detection, there are generally three steps: signal recognition, signal amplification, and signal output. There is no signal amplification process in these methods, which may lead to insufficient detection sensitivity. At the same time, the signal recognition procedure is not stable enough and is easily affected by interference factors, especially the non-target metal ion.

Shang designed a novel single universal primer multiplex ligation-dependent probe amplification (SUP-MLPA) technique [52]. The key principle is that only when the probes are exactly matched with the target template sequence can the two be connected for hybridization and amplification; even if there is a base difference, the amplification cannot be realized, which greatly improves the reaction specificity.

In this study, we designed a label-free DNAzyme-ligation-PCR-based colorimetric biosensor for Cu^2+^. We combined ligation-dependent probe amplification technology with DNAzyme. Then, the primers for hairpin DNA were designed, which include nucleic acid segments, blockers, and the G-rich sequence, for performing PCR to zip off of the hairpin structure, leading the G-rich sequence to fold to form a G-quadruplex structure. Thus, the formation of G-quadruplex was blocked by the addition of copper ions, resulting in signal reduction and the development of a fast, sensitive, and stable “turn-off” colorimetric detection method.

## 2. Materials and Methods

### 2.1. Materials and Reagents

The substrate sequence of Cu^2+^-dependent DNAzyme (Cu-Sub) and the enzyme sequence of Cu^2+^-dependent DNAzyme (Cu-Enz) and all primers were synthesized by Shanghai Sangon Co. Ltd. (Shanghai, China). The details of these primers and the nucleotide sequences are listed in Table 1. Ultrapure water used throughout all experiments was purified with a Milli-Q system (resistivity > 18.0 MΩ cm^−1^). The used metal salts, including Al(NO_3_)_3_, Li_2_SO_4_, Pb(NO_3_)_2_, Zn(NO_3_)_2_, KCl, CaCl_2_, CuCl_2_, FeCl_3_, MgCl_2_, MnCl_2_, NaCl, and FeCl_2_, were bought from the Sigma Chemical Company (St. Louis, MO, USA). Other reagents such as NaOH, H_2_O_2_ (30%), ABTS, Tris, HCl, hemin, 4-(2-hydroxyethyl) piperazine-1-ethanesulfonic acid (HEPES), sodium ascorbate, dimethyl sulfoxide (DMSO), and Trition X-100 were obtained from Baoxin Biotechnology Co. Ltd. (Kuning, China). All the chemical reagents were of analytical grade. The absorption spectrum of the reaction product was measured and recorded by a SpectraMax M5 plate reader (Molecular Devices, Sunnyvale, Calif.).

### 2.2. Preparation of DNAzyme

To form the Cu^2+^-dependent DNAzyme, Cu-Sub (10 μM) and Cu-Enz (10 μM) were mixed at a ration of 1:10 in a 200 μL buffer (1.5 M NaCl, 50 mM HEPES, pH 7.0). The mixture was warmed to 90 °C for 3 min and subsequently cooled naturally at room temperature for 1 h to anneal the DNAzyme. The resultant DNAzymes were stored at 4 °C.

### 2.3. Hybridization and Ligation

The hybridization and ligation reactions were conducted in 500 μL PCR reaction vessels using a thermocycler (Applied Biosystems). The 10× Ampligase reaction buffer, 2 U *Ampligase* (Epicentre, Madison, WI, USA), 1 μL of one of the probes (1 μM), and a 1.5 μL template were used as reaction mixes (5 μL total volume).

The conditions were as follows: the DNA was initially denatured at 95 °C for 3 min, followed by 20 denaturation and ligation cycles at 95 °C for 15 s and 65 °C for 5 min. After ligation, the reaction mixes were heated for 5 min at 98 °C to inactivate the enzyme and then stored in a refrigerator at 4 °C.

### 2.4. PCR Amplification

The reaction mixes used for ligation product amplification were prepared in a 25 μL total volume containing 1 μL of each primer (10 μM), 2.5 μL 10× Ex Taq buffer, 2 μL dNTP, 0.2 μL Ex Taq polymerase (TaKaRa Biotechnology Co. Ltd., Dalian, China), a 2 μL template from the ligation mix, and 16.3 μL of ultrapure water. DNA denaturation and polymerase activation at 95 °C for 5 min were followed by 30 amplification cycles at 95 °C for 30 s, 60 °C for 30 s, and 72 °C for 30 s, with a final step of 10 min at 72 °C. Finally, the reaction products were then cooled to 4 °C.

### 2.5. Colorimetric Determination of Cu^2+^

For each experiment, the DNA mixture (100 μL) containing 85 μL of ultrapure water, 10 μL of Cu^2+^ stock solution, 5 μL of DNAzyme, and 5 μL of sodium ascorbate (the final concentration was 5 mM) was sequentially added into a centrifuge tube. After 30 min of reaction at 25 °C, 1.5 μL of the above mixture was used as the template for the hybridization and ligation reactions (5 μL total volume). Next, during PCR amplification, the aim is to properly fold to form a large number of G-quadruplexes. Then, 2 μL hemin (final concentration of 2 μM) solution was added into the mixture system, mixed well, and incubated at room temperature for 1 h. Subsequently, 8 μL of ABTS (40 mM), 5 μL of H_2_O_2_ (20 mM), and 60 μL of buffer (100 mM HEPES, pH 7.0, 0.008% (*v*/*v*) Trition X-100, and 400 mM NaCl) were added. The final volume of the mixture was 100 μL, and the mixture was transferred into a 96-well microtiter plate. After 5 min, Spectra Max M5 (Molecular Devices, Sunnyvale, Calif.) was used to measure and record the absorbance of the reaction product at 419 nM, and the absorbance was used for subsequent quantitative analysis.

### 2.6. Analysis of Copper Ion Content in Yunnan Water Samples

Several kinds of water samples (Dianchi Lake and Cuihu Lake) were collected in Kunming, Yunnan Province, P.R. China, and prepared by mixing drinking water with standard Cu^2+^ solution. The concentrations of Cu^2+^ in the water samples were determined by the above-mentioned assay procedures. Conditions: The samples (10 μL) were added to a 100 μL detection system (containing DNAzyme solution and sodium ascorbate). Subsequently, we determined the samples according to the above-mentioned procedures, and the absorbance at 419 nm was recorded. In order to verify the results of this detection method, inductively coupled plasma-mass spectrometry (ICP-MS) was used to detect these water samples.

## 3. Results and Discussion

### 3.1. Principle of the Biosensor System

The principle of the DNAzyme-ligation-PCR-based visual “turn-off” colorimetric biosensor for Cu^2+^ is shown in Figure 1 (Appendix A). The system was a combination of three components. Firstly, the Cu^2+^-dependent DNAzyme was formed by the hybridization between the improved substrate strand and the enzyme strand. Two ligation-F/R probes were designed that were complementary to the substrate strand. Without Cu^2+^, the substrate strand of the DNAzyme complex was not cleaved. Under the ligase (*Ampligase*) and certain conditions, a ligation-based reaction occurred to produce ligation products. Conversely, if Cu^2+^ presents in the detection system, the substrate strand will be cut into two short chains at the recognition site, and the ligation-based reaction will not happen (Figure 1A).

Then, the sequences of hairpin DNA-1F/1R (H_1_/H_2_) were designed. The H_1_ includes a nucleic acid segment complementary to a downstream sequence of ligation products and is connected to a nucleic acid sequence and an oxyethyleneglycol bridge (blocker) to the nucleic acid sequence that is rich in G bases. The H_2_ includes a nucleic acid segment complementary to the template, a sequence linked through a blocker tether to the nucleic acid sequence rich in G bases. The blocker is important because the polymerization of the respective templates is always terminated at the blocker. After PCR amplification, the zipping off of the hairpin structure led the G-rich sequence to fold into a G-quadruplex (Figure 1B).

At last, hemin can bind to the formed G-quadruplex structure to yield a G-quadruplex DNAzyme with catalytic activity, resulting in an increase in the absorbance at 419 nm in the ABTS-H_2_O_2_ system (Figure 1C).

While Cu^2+^ was present in the sample, the substrate strand of Cu^2+^-dependent DNAzyme was cleaved, a ligation-based reaction was not occurring, and it could not form ligation products. Hence, there could not be a large number of active G-quadruplex DNAzymes, resulting in the signal reduction. Thus, a “turn-off” colorimetric detection of copper ions was achieved.

### 3.2. Verification of the Biosensor System

In order to verify the feasibility of the principle of the biosensor system, the primers UP-F/UP-R were designed. In Figure 1A, without Cu^2+^, the substrate strand and DNAzyme were present in the system, respectively, and the desired PCR products were analyzed through electrophoresis in a 2.0% agarose gel. While in the presence of Cu^2+^ and DNAzyme at the same time, the substrate strand was cleaved, as shown in Figure 1B. Our results suggested that the principle of the biosensor system is feasible.

### 3.3. Optimization of the Biosensor

In order to obtain the best detection system, the optimization of various conditions is necessary. The length of the incubation time of copper ions and DNAzyme has a significant effect on cutting efficiency, and 0.5 h, 1 h, and 1.5 h were set for the reaction time. The result (Figure 2A) shows that when time reaches 0.5 h, it can be completely cut. Thus, considering the time savings, 0.5 h is selected as the optimal cutting time.

The sensitivity depended on the ligation probe concentration; we set the initial concentration of the ligation probe at 10 μM and diluted tenfold from 10 to 0.0001 μM to obtain the amplifying target product proportional to the concentration. (Figure 2B), at 1 μM, amplified bands could be observed most clearly. Therefore, we selected 1 μM as the best probe concentration.

Under different conditions, the cutting efficiency of copper ions is also different. Three kinds of buffer were chosen to perform experiments, including buffer 1 (1.5 M NaCl, 50 mM HEPES, pH 7.0), buffer 2 (100 mM HEPES, pH 7.0, 0.008% (*v*/*v*) Trition X-100, and 400 mM NaCl), and buffer 3 (ultrapure water). The result is shown in Figure 2C. Under the condition of buffer 3, the change in absorbance (ΔAbs, ΔAbs = Abs0 − Abs, where Abs0 represents the absorbance in the absence of copper ions and Abs represents the absorbance in the presence of copper ions) was the biggest. Thus, buffer 3 was used in this study.

The ratio of Cu–Enz and Cu–Sub will affect the amount of substrate cleavage. In order to achieve consistent amplification results, the ratio of Cu-Enz to Cu-Sub was optimized by fixing the substrate chain concentration and changing the amount of Cu-Enz (1:1, 3:1, 5:1, 10:1, and 15:1). Results according to the different ratios of Cu-Enz to Cu-Sub (Figure 2D) at 10:1 and 15:1, the effect is basically the same. Therefore, considering the lower cost, the ratio of Cu-Enz to Cu-Sub was 10:1.

When the concentration of hydrogen peroxide (H_2_O_2_) is too low, its oxidation ability is not strong; on the contrary, if the concentration of H_2_O_2_ is too high, it will strengthen the disproportionation. Therefore, it is necessary to optimize the concentration of H_2_O_2_. When the final concentration of H_2_O_2_ reached 1 mM, the effect of the experiment was the best (Figure 2E).

Under different ionic conditions, G-quadruplex, which binds to hemin, showed different catalase activity. The experiments were carried out by selecting six different conditions (buffers 1–6, in Figure 2F). The results (Figure 2F) showed that the best catalytic activity was obtained under the condition of buffer 2. The original measurement data of Figure 2C, Figure 2D, Figure 2E and Figure 2F are shown in Appendix A respectively.

### 3.4. Sensitivity of the Biosensor

Applying the above optimization conditions, the sensitivity of the biosensor was determined by recording the absorption signal after the addition of different concentrations of Cu^2+^. As shown in Figure 3A, with the increase in the concentration of copper ions, the absorbance signal decreased gradually until reaching a plateau. A linear relationship was observed in Figure 3B; the linear regression equation was Y = −0.13253X + 0.545 (R^2^ = 0.994) over a range of 31.25 nM to 500 nM Cu^2+^, and the detection limit was determined to be 18.25 nM (3σ/slope, where σ is the standard deviation for determining the blank sample), which represents one of the most sensitive Cu^2+^ determination methods. According to the World Health Organization (WHO), which defined the maximum permitted contamination level of Cu^2+^ in drinking water to be 20 μM, this detection limit is far below the regulated level. The original measurement data in Figure 3B had shown in Appendix A.

### 3.5. Selectivity of the Biosensor

To evaluate the selectivity of the biosensor to Cu^2+^, different relevant metal ions with a concentration of 10 μM were used for detection, including Cu^2+^, Fe^3+^, Fe^2+^, Al^3+^, Ca^2+^, Pb^2+^, Mn^2+^, Zn^2+^, Li^+^, and Mg^2+^. In Figure 4A,B, in the presence of Cu^2+^, the change of absorbance (ΔAbs, ΔAbs = Abs0 − Abs, where Abs0 represents the absorbance in the absence of metal ions and Abs represents the absorbance in the presence of metal ions) gave a remarkable enhanced absorbance signal and had a noticeable weakening of the color. These results indicated that the biosensor system had good selectivity toward Cu^2+^. The original measurement data in Figure 4A had shown in Appendix A.

### 3.6. Application of the Biosensor

To investigate the practical application and the analytical reliability of the biosensor, combined with the geographical environment, we selected two water samples from Dianchi Lake and Cuihu Lake in Yunnan Province as representative samples. The method developed in this study and a highly sensitive, classical metal ion detection method (inductively coupled plasma-mass spectrometry, ICP-MS) were used for standard addition recovery experiments, and the detection results were compared and analyzed.

In this experiment, the copper ion content of the sample itself was first detected by two methods, and then the copper ion with a concentration of 0.1 μM was added to the sample for mixing and detection again. The detection values of the two methods were compared, and the recovery rate was calculated to verify the practicability of the detection method in this study. The experimental results are shown in Table 2. The original measurement data in Table 2 had shown in Appendix A.

For the selection of test samples, this study considered that Dianchi Lake is the largest fresh lake in Yunnan [53], which is located in the enrichment area of phosphorus [54], and it has been listed as one of the three most polluted lakes in China [55]. Cuihu Lake is located in the center of the main urban area of Kunming, with vigorous commercial development [56]. Its social and economic development has made its water pollution increasingly serious. Due to some natural factors, as well as agricultural fertilizer water, industrial wastewater, and the inflow of domestic sewage from surrounding residents, these lakes have eutrophication (water bodies rich in nitrogen, phosphorus, and other substances) [57], and there is a problem of heavy metal ion pollution [58]. Compared with other water sources (such as tap water or rainwater), the composition of the lake sample itself was more complex and had a greater impact on the measurement results of the detection method. Therefore, these two different polluted lake water samples were selected as test samples, which can more effectively illustrate the applicability of this research method.

It can be seen from Table 2 that the concentrations of copper ions in Dianchi Lake and Cuihu Lake measured by this method are 0.22 μM and 0.179 μM, respectively, and the concentrations measured by ICP-MS are 0.228 μM and 0.184 μM, respectively. The difference between the results measured by these two methods is very small; similarly, the recovery rates of copper ion concentration in Dianchi Lake and Cuihu Lake samples detected by this method were 94–96% and the RSD was 4–7%; the recovery rates detected by ICP-MS were 89–97% and the RSD was 1–3%. These results show that the detection method has good reliability in practical applications.

Therefore, this method uses the above two representative polluted water samples as the actual sample detection and obtains good test results, which can show that the detection method has a wide range of applications and can be used for daily drinking water, agricultural irrigation water, domestic sewage, and other water detection.

In consideration of low cost, simple operation steps, and no need to use expensive large instruments, by comparing the detection limits of various methods for copper ions, it can be seen that the method of this study has an excellent detection limit (Table 3).

## 4. Conclusions

For the whole study, we designed a fast, sensitive, and stable DNAzyme-ligation-PCR-based visual “turn-off” colorimetric biosensor for Cu^2+^. In this system, we combined ligation-dependent probe amplification technology with DNAzyme for the first time. Then, the special primers were designed, which include nucleic acid segments, blockers, and G-rich sequences, for performing PCR to fold into G-quadruplexes without Cu^2+^. In this paper, some conditions of the biosensor system were optimized. The specificity and sensitivity of the system were measured under optimal conditions. The detection limit of the sensor is 18.25 nM, which is far below the regulated concentration (20 μM). Moreover, through the evaluation of different water samples, the results show that the biosensor system has good practicability and feasibility.

In addition to copper ions, this detection method can also be used for the detection of other heavy metal ions. For the detection limit and sensitivity of this study, they may be enhanced by using color enhancers to increase the G-quadruplex coloration and improved by using other signal amplification methods. At present, this method can only detect one kind of metal ion, and the realization of the detection method also involves a change in temperature. In the subsequent research, it can be developed in the direction of multiple metal ion detection with an isothermal procedure.

## Data Availability

The data presented in this study are available on request from the corresponding author.

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
