# Peer review of "A Novel Cu2+ Quantitative Detection Nucleic Acid Biosensors Based on DNAzyme and “Blocker” Beacon"

_foods, 2023, doi:10.3390/foods12071504_

Round 1
Reviewer 1 Report
The authors presented a novel Cu2+ quantitative detection nucleic acid biosensors based on DNAzyme and “blocker” beacon. The article is very organized and well written. The aim was very clear. Method optimization is excellent. However, many revisions are required.
1- The merits for the new method over old similar ones (Cu2+-dependent DNAzyme ) should be highlighted in the abstract
2- The following article should be cited in the introduction
https://pubs.acs.org/doi/abs/10.1021/acs.analchem.5b04904
add
3- Scheme 1 is very difficult to follow, you can keep it with the original size in supplementary file. The figures should be readable and understandable.
4- In introduction the following statement in lines 98-99; needs further details “However, these methods have no signal amplification process and have the problems of poor stability and easily affected by interference factors.”
5- In line 174, mention full name for ICP-MS
6- In line 197, add space a greatnumber
7- In lines 230-237, either use buffer c or buffer 3 for uniformity.
8- Hemin, short description for it is required.
9- Validation data was missed. Please provide it. May be in supplementary file
10- A table for comparison the detection limit for the new method and other old published Cu2+-dependent DNAzyme methods should be provided
11- Study limitation and future plan should be provided.
Best wishes
Author Response
Dear Editors and Reviewers:
Thank you for your letter again and for the reviewers’ comments concerning our manuscript entitled “A novel Cu2+ quantitative detection nucleic acid biosensors based on DNAzyme and “blocker” beacon”. Those comments are all valuable and very helpful for revising and improving our paper, as well as the important guiding significance to our researches. We have replied the comments carefully and have made correction which we hope meet with approval. Revised portion are highlighted in red in the manuscript. The main corrections in the paper and the responds to the reviewer’s comments are as following:
Reviewer #1: The author has revised the manuscript, but there are still some issues that need to be resolved:
- The merits for the new method over old similar ones (Cu2+-dependent DNAzyme ) should be highlighted in the abstract .
Response: Thank you very much for the comments and constructive suggestions. We have added the advantages of the new method over the old one in Abstract section. (Line 17-19)
- The following article should be cited in the introduction
https://pubs.acs.org/doi/abs/10.1021/acs.analchem.5b04904 add
Response: Thank you very much for the comments, we have added this excellent literature to the introduction section on copper ion detection. (Line 69, ref. No. 22)
- Scheme 1 is very difficult to follow, you can keep it with the original size in supplementary file. The figures should be readable and understandable.
Response: Thank you very much for the constructive suggestions. We appropriately adjusted the background color and specific content of scheme 1, and revised the descriptions to make scheme 1 easily to understand. The scheme 1 is presented in the original size in the supplementary material also. (Line 187-206)
- In introduction the following statement in lines 98-99; needs further details “However, these methods have no signal amplification process and have the problems of poor stability and easily affected by interference factors.”
Response: Thank you very much for the positive comments and constructive suggestions. We realized that the previous use of language is not appropriate, and re-summarized this part and made some detailed explanations. (Line 102-106)
- In line 174, mention full name for ICP-MS
Response: Thanks very much for your suggestion, we have added its full name where ICP-MS appeared. (Line 181)
- In line 197, add space a greatnumber
Response: Thanks very much for your suggestion, we have added a space in the suitable location. (Line 209)
- In lines 230-237, either use buffer c or buffer 3 for uniformity.
Response: Thanks very much for your suggestion, we used buffer 3 for uniformity. (Line 241-247)
- Hemin, short description for it is required.
Response: Thanks very much for your suggestion, we have added the corresponding description of the role of hemin in text. (Line 91-92)
- Validation data was missed. Please provide it. May be in supplementary file
Response: Thanks very much for your suggestion, we provided all data processing prior to initial measurement results in the file of “Supplementary Materials”.
- A table for comparison the detection limit for the new method and other old published Cu2+-dependent DNAzyme methods should be provided.
Response: Thanks very much for your suggestion, we have made a comparison between the detection limits of our detection method and other copper ion detection methods, and the comparison results had presented through the table. (Line 353-356, and Table 3)
- Study limitation and future plan should be provided.
Response: Thank you very much for the positive comments and constructive suggestions, we added
Study limitation and future plan in Conclusions. (Line 369-375)
These are our revisions and responses for your comments. We sincerely hope that this revised manuscript has addressed all your comments and suggestions. We appreciate the reviewers' enthusiastic work and hope that the corrections will be approved. Once again, thank you very much for your comments and for giving us the opportunity to resubmit. We are greatly looking forward to hearing from you.
Thank you and best regards.
Yours sincerely

Reviewer 2 Report
In this article highly sensitive and selective “turn-off” colorimetric sensor for monitoring of copper (II) ions in aqueous samples was developed. In my opinion the manuscript is interesting and well organized and it contains valuable information.
Minor comments:
As I understand correctly, the experiment for determination of sensor selectivity was carried out for all metal ions separately. Do the authors provide the cross selectivity experiment and investigate the response for copper(II) ions in the presence of interfering ions?
Why did the authors not investigated the response of the sensor to mercury(II) ions?
Author Response
Dear Editors and Reviewers:
Thank you for your letter again and for the reviewers’ comments concerning our manuscript entitled “A novel Cu2+ quantitative detection nucleic acid biosensors based on DNAzyme and “blocker” beacon”. Those comments are all valuable and very helpful for revising and improving our paper, as well as the important guiding significance to our researches. We have replied the comments carefully and have made correction which we hope meet with approval. Revised portion are highlighted in red in the manuscript. The main corrections in the paper and the responds to the reviewer’s comments are as following:
Responds to the reviewers' comments:
Reviewer #2: The author has revised the manuscript, but there are still some issues that need to be resolved:
1, As I understand correctly, the experiment for determination of sensor selectivity was carried out for all metal ions separately. Do the authors provide the cross selectivity experiment and investigate the response for copper (II) ions in the presence of interfering ions?
Response: Thank you very much for the comments and constructive suggestions. We did a simple cross-response experiment in the pre-experiment, and the results showed that the presence of other ions almost did not affect the response of the biosensor. In the formal experiment, we first carried out the selectivity of the biosensor experiment with a variety of metal ions, and the results showed that the biosensor system had good selectivity towards Cu2 +. Then in the actual sample detection, the lake water samples we selected had the characteristics of complex components, meanwhile, the final detection results were almost consistent with ICP-MS, which indicated that our sensor detection method had certain anti-interference ability. And experiments have shown that this group of copper enzyme chains and substrate chains also show good specificity in metal ion mixtures1, 2.
Reference
- Wu, J. K.; Yu, Y. L.; Wei, S. H.; Xue, B.; Zhang, J. L., A DNAzyme-based Electrochemical Impedance Biosensor for Highly Sensitive Detection of Cu2+ Ions in Aqueous Solution. International Journal of Electrochemical Science 2017, 12 (12), 11666-11676.
- Xu, W. T.; Tian, J. J.; Luo, Y. B.; Zhu, L. J.; Huang, K. L., A rapid and visual turn-off sensor for detecting copper (II) ion based on DNAzyme coupled with HCR-based HRP concatemers. Scientific Reports 2017, 7.
2, Why did the authors not investigated the response of the sensor to mercury (II) ions?
Response: Thank you very much for the comments and constructive suggestions. In the sensor selectivity experiment, we selected some heavy metal ions to do the specificity test. As it happens, Hg2+ were not selected at that time. In terms of enzyme chain and substrate chain, the enzyme chain and substrate chain of these two metal ions are different and the mechanism is also different. Some studies have developed detection methods based on DNAzyme for Cu2+ and Hg2+, and both have used Cu2+ and Hg2+ for specific detection. The results show that the two ions have little interference with the detection to each other3, 4.
Reference
- Wang, F.; Orbach, R.; Willner, I., Detection of Metal Ions (Cu2+, Hg2+) and Cocaine by Using Ligation DNAzyme Machinery. Chemistry-a European Journal 2012, 18 (50), 16030-16036.
- Li, H.; Huang, X. X.; Cai, Y.; Xiao, H. J.; Zhang, Q. F.; Kong, D. M., Label-Free Detection of Cu2+ and Hg2+ Ions Using Reconstructed Cu2+-Specific DNAzyme and G-quadruplex DNAzyme. Plos One 2013, 8 (9).
These are our revisions and responses for your comments. We sincerely hope that this revised manuscript has addressed all your comments and suggestions. We appreciate the reviewers' enthusiastic work and hope that the corrections will be approved. Once again, thank you very much for your comments and for giving us the opportunity to resubmit. We are greatly looking forward to hearing from you.
Thank you and best regards.
Yours sincerely

Reviewer 3 Report
Comments to the authors:
1. The specificity test of the biosensor should be carried out.
2. The comparison table should be provided and the similar studies should be included.
3. Scheme 1 is so vague. it is better to change the background and provide another scheme.
4. RSD of the biosensor and ICP-MS should be considered.
Author Response
Dear Editors and Reviewers:
Thank you for your letter again and for the reviewers’ comments concerning our manuscript entitled “A novel Cu2+ quantitative detection nucleic acid biosensors based on DNAzyme and “blocker” beacon”. Those comments are all valuable and very helpful for revising and improving our paper, as well as the important guiding significance to our researches. We have replied the comments carefully and have made correction which we hope meet with approval. Revised portion are highlighted in red in the manuscript. The main corrections in the paper and the responds to the reviewer’s comments are as following:
Responds to the reviewers' comments:
Reviewer #3: The author has revised the manuscript, but there are still some issues that need to be resolved:
- The specificity test of the biosensor should be carried out.
Response: Thank you very much for the comments and constructive suggestions. We have evaluated the selectivity of the biosensor and tested the specificity of the sensor with a variety of metal ions, the results show that our sensor has good specificity for copper ions (section 3.4 and 3.5). In addition, in the actual sample detection, we used lake samples with complex compositions, which contained many interfering substances. Compared with the traditional instrument detection method ICP-MS, the detection result of our method is also good, which indicates that our detection method has good specificity.
- The comparison table should be provided and the similar studies should be included.
Response: Thank you very much for the comments and constructive suggestions. We lacked a comparison of detection limits with other detection methods, so we made a comparison between the detection limits of our detection method and those of other copper ion detection methods, and the comparison results will be presented through the Table 3. (Line 353-356)
- Scheme 1 is so vague. it is better to change the background and provide another scheme.
Response: Thank you very much for the constructive suggestions. We appropriately adjusted the background color and specific content of scheme 1, and revised the descriptions to make scheme 1 easily to understand. The scheme 1 is presented in the original size in the supplementary material also. (Line 187-206)
- RSD of the biosensor and ICP-MS should be considered.
Response: Thank you very much for the comments and constructive suggestions. We have supplemented these figures in the Table S7.
These are our revisions and responses for your comments. We sincerely hope that this revised manuscript has addressed all your comments and suggestions. We appreciate the reviewers' enthusiastic work and hope that the corrections will be approved. Once again, thank you very much for your comments and for giving us the opportunity to resubmit. We are greatly looking forward to hearing from you.
Thank you and best regards.
Yours sincerely

Reviewer 4 Report
The manuscript of Zhang et al. entitled “A novel Cu2+ quantitative detection nucleic acid biosensors based on DNAzyme and “blocker beacon” describes the development of colorimetric biosensor for detection of copper (II) ion based on Cu2+-dependent DNAzyme and “blocker” beacon. Elaborated device exhibited a linear calibration from 0.03125 to 0.5 μM and the limit of detection of 18.25 nM. Additionally, successful confirmation of the selectivity of the system was performed by testing different relevant metal ions. Furthermore, this copper (II) ion biosensor was effectively applied to monitor Cu2+ in real water samples.
Therefore I recommend publication of the manuscript after minor revision:
1. Error bars should be included in Fig. 2C-F, Fig. 3B, C and Fig. 4A. How many repetitions have been performed?
2. Lack of Y axis caption in Fig. 3A.
3. There is no sub-point C in the caption of Fig. 3A.
4. A comparison of the performance of the proposed sensing platform with other reported systems designed for detection of the copper (II) ion should be added. In addition, presenting the information in a table format makes the data much easier to follow.

Author Response
Dear Editors and Reviewers:
Thank you for your letter again and for the reviewers’ comments concerning our manuscript entitled “A novel Cu2+ quantitative detection nucleic acid biosensors based on DNAzyme and “blocker” beacon”. Those comments are all valuable and very helpful for revising and improving our paper, as well as the important guiding significance to our researches. We have replied the comments carefully and have made correction which we hope meet with approval. Revised portion are highlighted in red in the manuscript. The main corrections in the paper and the responds to the reviewer’s comments are as following:
Responds to the reviewers' comments:
Reviewer #4: The author has revised the manuscript, but there are still some issues that need to be resolved:
- Error bars should be included in Fig. 2C-F, Fig. 3B, C and Fig. 4A. How many repetitions have been performed?
Response: Thank you very much for the comments and constructive suggestions. We have measured three parallels in each measurement. Due to the negligence of our picture making, we have not shown the error situation. We have re-added the error bar on each required picture, and also put the raw data in the file of “Supplemental Materials”.
- Lack of Y axis caption in Fig. 3A.
Response: Thank you very much for the suggestion. We have added the caption of Y axis in Fig. 3A.
- There is no sub-point C in the caption of Fig. 3A.
Response: Thank you very much for the suggestion. Because of our negligence, we lost the title and related narrative of Fig.3C, and now we have added it (Line 289-296).
- A comparison of the performance of the proposed sensing platform with other reported systems designed for detection of the copper (II) ion should be added. In addition, presenting the information in a table format makes the data much easier to follow.
Response: Thanks very much for your suggestion, we have made a comparison between the detection limits of our detection method and other copper ion detection methods, and the comparison results had presented through the table. (Line 353-356, and Table 3)
These are our revisions and responses for your comments. We sincerely hope that this revised manuscript has addressed all your comments and suggestions. We appreciate the reviewers' enthusiastic work and hope that the corrections will be approved. Once again, thank you very much for your comments and for giving us the opportunity to resubmit. We are greatly looking forward to hearing from you.
Thank you and best regards.
Yours sincerely

Round 2
Reviewer 1 Report
the authors replied in a very professional way. I appreciate their efforts. the paper could be accepted in the current form.
Greetings
Author Response
Dear Reviewer,
Thank you for your comments on our mansucript, which are very important for us to improve the level of our mansucript, Thank you again for your affirmation of our reply.
best regards
Yours sincerely
Reviewer 3 Report
The most concerns have been addressed. The schematic is still unclear. RSD should be added in Table. 2.
Author Response
Dear Editors and Reviewers:
Thank you for your letter again and for the reviewers’ comments concerning our manuscript entitled “A novel Cu2+ quantitative detection nucleic acid biosensors based on DNAzyme and “blocker” beacon”. Those comments are all valuable and very helpful for revising and improving our paper, as well as the important guiding significance to our researches. We have replied the comments carefully and have made correction which we hope meet with approval. Revised portion are highlighted in red in the manuscript. The main corrections in the paper and the responds to the reviewer’s comments are as following:
Responds to the reviewers' comments:
Reviewer #3: The author has revised the manuscript, but there are still some issues that need to be resolved:
- The schematic is still unclear.
Response: Thank you very much for the comments and constructive suggestions. We adjusted the schematic again to make it more clearer. The scheme 1 was presented in the original size in the supplementary material also.
- The schematic is still unclear. RSD should be added in Table. 2.
Response: Thank you very much for the comments and constructive suggestions. The RSD have been added to Table 2. (Line 322-323)
These are our revisions and responses for your comments. We sincerely hope that this revised manuscript has addressed all your comments and suggestions. We appreciate the reviewers' enthusiastic work and hope that the corrections will be approved. Once again, thank you very much for your comments and for giving us the opportunity to resubmit. We are greatly looking forward to hearing from you.
Thank you and best regards.
Yours sincerely
